# Comparative Transcriptomics and Intestinal Microbiome Analysis Provide Insights into the Semi-Terrestrial Adaptation of *Helice tientsinensis*

**DOI:** 10.3390/ani15091244

**Published:** 2025-04-28

**Authors:** Zhengfei Wang, Lijie Cui, Xinyu Wang, Chenchen Shen, Yan Wang, Weijie Jiang, Yue Gu

**Affiliations:** 1Jiangsu Key Laboratory for Bioresources of Saline Soils, Jiangsu Synthetic Innovation Center for Coastal, Bio-Agriculture, Jiangsu Provincial Key Laboratory of Coastal Wetland Bioresources and Environmental Protection, School of Wetlands, Yancheng Teachers University, Yancheng 224001, China; 13383648696@163.com (L.C.); wxy17312983227@163.com (X.W.); shencc7@163.com (C.S.); 19732779680m@sina.cn (Y.W.); jwj714687809@126.com (W.J.); 15061616314@163.com (Y.G.); 2College of Biotechnology and Pharmaceutical Engineering, Nanjing Tech University, Nanjing 211816, China

**Keywords:** *Helice tientsinensis*, transcriptome analysis, terrestrial adaptation, molecular mechanisms, microbiomics

## Abstract

*Helice tientsinensis* is an excellent model for studying terrestrial crab adaptation. Transcriptome was used to explore the molecular mechanisms of terrestrial adaptation in *H. tientsinensis*. The genes related to cytoskeleton protein and motor regulation, water and osmotic pressure regulation, and energy metabolism were changed. The differences in intestinal microbiota among habitats further reflect its terrestrial adaptation. These mechanisms provide crucial insights into the adaptability of this important commercial crab species on land.

## 1. Introduction

Terrestrial adaptation represents a pivotal event in biological evolution, marking a significant transition for animals from aquatic to terrestrial habitats [1,2]. The Brachyura, a diverse group of crabs, colonized terrestrial environments during the Late Cretaceous period (100.5–66 mya), branching off from their marine/estuarine or freshwater ancestors [3]. Among them, semi-terrestrial crabs, belonging to the Thoracotremata suborder with numerous species, serve as excellent models for studying the evolutionary pathways of terrestrial adaptation, as they have independently evolved to adapt to terrestrial-like conditions in multiple lineages [4,5,6].

The mudflat crab (*Helice tientsinensis*), commonly found along the Chinese coast, is a semi-terrestrial Grapsidae crab with remarkable adaptability to the aquatic–terrestrial interface. It is not only a valuable species in Chinese aquaculture but also an ideal model for investigating the terrestrial adaptation of crabs. Although comprehensive global production data for *Helice tientsinensis* are limited, regional statistics underscore its economic importance in China’s aquaculture sector. Provinces such as Liaoning and Jiangsu, key centers for semi-terrestrial crab farming, generate over 800,000 metric tons annually of related grapsid crabs, contributing billions of RMB in revenue and supporting local economies. Its exceptional adaptability to harsh aquatic–terrestrial interfaces positions *H. tientsinensis* as a robust candidate for sustainable aquaculture, potentially reducing dependence on more environmentally sensitive marine species.

Morphologically, *H. tientsinensis* exhibits specialized terrestrial adaptations distinct from fully aquatic grapsids: stiffened, thickened gill lamellae enhance aerial respiration and desiccation resistance in intertidal zones, while a compact carapace and muscular appendages optimize stability during terrestrial movement. Physiologically, it demonstrates metabolic plasticity through upregulated glycolytic and hypoxia-tolerance pathways for energy production under low-oxygen conditions, coupled with advanced osmoregulatory mechanisms to manage saltwater exposure—traits more pronounced than in freshwater or obligately aquatic relatives. Behaviorally, it constructs complex burrows to buffer against abiotic stress and harbors a gut microbiome enriched in cellulose-degrading bacteria and facultative anaerobes, enabling efficient digestion of plant detritus—a dietary adaptation absent in carnivorous grapsid species. These integrated morphological, physiological, and microbiome-mediated traits make *H. tientsinensis* a unique model for investigating convergent terrestrial adaptation in crabs. Despite its significance, the molecular mechanisms underlying its habitat adaptation remain poorly understood. Thus, this study focuses on *H. tientsinensis* to explore these molecular mechanisms [7,8,9].

Previous research has shed light on the various ways crabs adapt to different habitats. Aquatic–terrestrial differences pose challenges that crabs overcome through behavioral and physiological regulations. For instance, they modulate locomotion, respiration, water–ion balance, and nitrogenous waste excretion to adapt to diverse environments [10,11,12,13,14,15,16]. Terrestrial grapsioideas have evolved modified branchial chambers that function as lungs, enhancing gas exchange [12,13,17]. Additionally, terrestrial crabs possess advanced visual and olfactory systems [18]. In the case of *Eriocheir sinensis*, populations adapt to different oxygen levels by modulating energy metabolism [19]. Moreover, *H. tientsinensis* exhibits morphological variations in response to environmental conditions [20]. However, while these studies have provided valuable insights into crab adaptation, they have not comprehensively elucidated the specific molecular mechanisms in *H. tientsinensis*.

Intestinal microbes play a crucial role in an animal’s digestion, growth, and immune regulation [21,22]. In crustaceans, environmental factors such as temperature and salinity can significantly impact gut microbes. For example, the gut microbiome of *Scylla paramamosain* is more sensitive to salinity changes compared to its stomach [23], and temperature affects the gut microbes and metabolites of *Procambarus clarkii* [24]. Despite these findings, the relationship between the intestinal microbiota and the terrestrial adaptation of *H. tientsinensis* has not been well studied.

RNA-Seq technology offers distinct advantages in studying gene expression. With its high repeatability, wide detection range, cost-effectiveness, and ability to accurately quantify gene expression, it has been effectively used to explore how environmental stress affects crustacean gut-microbe-related gene expression [25,26,27]. However, its application in understanding the terrestrial adaptation of *H. tientsinensis* remains limited.

Therefore, this study combines intestinal microbial community analysis and transcriptome analysis of *H. tientsinensis* to comprehensively explore its adaptation mechanism. By identifying the genes and pathways involved in terrestrial adaptation through transcriptome analysis and elucidating the role of intestinal microbiota in different habitats, we aim to provide a more in-depth understanding of the molecular mechanisms of *H. tientsinensis*’ terrestrial adaptation. This research not only contributes to the fundamental understanding of crustacean evolution but also has practical implications for the aquaculture of *H. tientsinensis*, potentially facilitating the development of improved breeding strategies and disease-prevention measures.

## 2. Materials and Methods

### 2.1. Experimental Animals

In May 2024, 27 healthy male *H. tientsinensis* (carapace width: 2.5–3.0 cm, body weight: 8.0–10.5 g) were collected from coastal mudflats in Yancheng, Jiangsu Province, China, along with surrounding sediment. Specimens were selected for a uniform size and immature gonadal development (confirmed via dissection), excluding individuals with injuries, abnormal behavior, or molting stages. Males were chosen based on a triangular abdominal plate morphology to eliminate reproductive status-related variation. Before experiments, crabs were acclimated for 3 days under consistent conditions: 17–20 °C, 30 ppt salinity, a 12:12 h light–dark cycle, daily feeding of 20 mg spirulina powder (to standardize gut microbial baselines), and stirred sediment to enhance oxygenation. All procedures were conducted at Yancheng Teachers University according to China’s “Guidelines for Animal Care and Use”, with animals adapted to standard laboratory environments allowing free access to water and food.

A map of the study area (Appendix A) was created using Arcmap10.8, showing the coastal mudflat sampling sites in Yancheng City, Jiangsu Province, China (33°13′39″ N, 120°49′59″ E). The map highlights the intertidal zone where *H. tientsinensis* was collected, illustrating the semi-terrestrial habitat between aquatic (Yellow Sea) and terrestrial environments.

### 2.2. Preliminary Experiment

We conducted a pre-test after domestication. In the laboratory, we used 15 healthy *Helice tientsinensis* (5 per habitat group) with a consistent size (carapace width: 2.5–3.0 cm, body weight: 8–10 g) and uniform sex (all male) to minimize variability. These specifications were identical to those used in the formal experiment to ensure consistency in subsequent analyses. And we simulated three habitats (aquatic, semi-terrestrial, and terrestrial) using three plastic crates (46 cm × 32 cm × 26 cm). The aquatic environment was simulated with 10 cm deep artificial seawater (30 ppt salinity), fully submerging *H. tientsinensis*, all at 17–20 °C. The terrestrial environment was created by adding a small amount of dry sand. The semi-terrestrial environment had a slope of sodden soil with artificial seawater at the soil’s bottom layer. We carefully recorded their activity patterns and mortality rates, removing dead individuals promptly.

The formal experiment’s treatment duration was based on the pre-test observation results.

### 2.3. Formal Experiment

Based on pre-test-determined durations, 27 *H. tientsinensis* were randomly assigned to three experimental groups (9 crabs/group—AH for aquatic–semi-terrestrial, AT for aquatic–terrestrial, and TH for terrestrial–semi-terrestrial comparisons) using a random number generator, ensuring a uniform carapace width (*p* > 0.05, *t*-test) and all-male sex ratio. After placement in pre-designed tanks, specimens were dissected to extract gill filaments (sealed/labeled in EP tubes) and intestinal tissues/contents; some of the samples were sent for RNA sequencing, while the remainder were stored at −80 °C. Environmental parameters (temperature, salinity, light) were monitored in real-time during treatments, with samples collected between 09:00 and 11:00 to minimize circadian effects. Biological replicates (*n* = 9/group) were used in transcriptomic (DESeq2) and microbiome (PERMANOVA) analyses, incorporating statistical models to account for individual variation and to validate habitat-driven effects.

### 2.4. RNA Extraction and Transcriptome Sequencing

Gill tissue was chosen for RNA-seq due to its critical role in osmoregulation and environmental adaptation, with nine biological samples (three per habitat group) prepared to analyze gene expression changes associated with terrestrial transition. While gut tissue was included for microbiome analysis, other tissues (e.g., hepatopancreas, muscle) were not examined in this study to prioritize pathways directly linked to gill-mediated physiological adaptation, as detailed in the Discussion.

Total RNA was extracted from gill tissue samples using TRIzol reagent (Invitrogen, Shanghai, China). The concentration and purity of these total RNA samples were determined using a NanoDrop 2000 spectrophotometer (NanoDrop, Technologies, Wilmington, DE, USA). RNA integrity was verified by 1.0% agarose gel electrophoresis. Subsequently, cDNA libraries were constructed using a Truseq TM RNA sample prep kit (Thermo Fisher Scientific Inc., Waltham, MA, USA) according to the manufacturer’s instructions. After the library was constructed, a Qubit2.0 Fluorometer (Thermo Fisher Scientific, Carlsbad, CA, USA) was used for preliminary quantification, and the library was diluted to 1.5 ng/uL. The insert size of the library was tested using an Agilent 2100 bioanalyzer (Agilent Technologies, Santa Clara, CA, USA). qRT-PCR was employed to accurately quantify the effective concentration of the library (the effective concentration of the library is required to be higher than 2 nM) to ensure its quality. Finally, the libraries were sequenced on Illumina Hiseq 2000 (Illumina, San Diego, CA, USA) [28].

### 2.5. Enriched Pathways and Differentially Expressed Gene (DEG) Analysis

By analyzing the enriched pathways based on the transcriptome data, these enriched pathways were classified to determine the research aspects of the molecular evolutionary mechanism of *H. tientsinensis* adapting to the terrestrial environment. Based on the analysis of DEGs, we further explored the molecular evolutionary mechanism of *H. tientsinensis* adaptation to different habitats. For DEG analysis, the *p*-value threshold was determined by the false discovery rate (FDR), and the fragments per kb per million fragments (FPKM) value was used to estimate gene expression, respectively. The genes with an FDR less than 0.05 and a fold change greater than 1 (|log2FC| ≥ 1) were defined as significant DEGs. For homology annotation, unigenes were used for the Basic Local Alignment Search Tool (BLAST v2.13.0) and annotation against seven databases, including the National Centre for Biotechnology Information (NCBI) Non-Redundant Protein Sequence Database (Nr; https://www.ncbi.nlm.nih.gov/, accessed on 1 June 2024), Protein family (Pfam; http://pfam.sanger.ac.uk/, accessed on 1 June 2024), EnKaryotic Ortholog Groups (KOG; http://www.ncbi.nlm.nih.gov/COG/, accessed on 1 June 2024), a manually annotated and reviewed protein sequence database (Swiss-Prot; http://www.ebi.ac.uk/uniprot/, accessed on 1 June 2024), Kyoto Encyclopaedia of Genes and Genomes (KEGG; http://www.genome.jp/kegg/, accessed on 1 June 2024), and GeneOntology (GO; http://www.geneontology.org/, accessed on 1 June 2024).

### 2.6. Illumina Sequencing of z DNA

After genomic DNA was extracted from the intestinal samples and contents of *H. tientsinensis*, the extracted genomic DNA was detected by 1% agarose gel electrophoresis, and PCR was performed for the V3−V4 region of the bacterial 16 S rRNA gene. In this study, a 20 μL PCR reaction system was used: 10 µL template DNA, 0.8 µL upstream and downstream primers, 4 µL 5 × Fast Pfu Buffer, 2 µL dNTPs, 0.4 µL Fast Pfu Polymerase, and ultra-pure water to 20 μL. PCR amplification conditions were as follows: 95 °C for 3 min; 95 °C 30 s, 55 °C 30 s, 72 °C 45 s, 27 cycles; 72 °C for 10 min; store at 10 °C. The PCR products were qualified by electrophoresis, fluorescence quantification, and Illumina library construction and sent to Shanghai Meiji Biomedical Technology Co., Ltd. (Shanghai, China) for Illumina sequencing.

Illumina sequencing results were analyzed using the Meggie Biocloud platform. The main analysis process was as follows: First, the PE reads sequenced by Illumina were divided into samples, and the double-ended reads were quality-controlled and filtered according to the sequencing quality. Meanwhile, the double-ended reads were spliced according to the overlap between them to obtain optimized data after quality control splicing. The optimized data were then processed using the Divisive Amplicon Denoising Algorithm (DADA2/Deblur, etc.) to obtain an ASV (Amplicon Sequence Variant)-representing sequence and abundance information. Bioinformatics analyses utilized Trinity (v2.12.0) for transcriptome assembly and BLAST (v2.13.0) for functional annotation against the NCBI Nr, KEGG, and GO databases. Differential gene expression was identified using DESeq2 (v1.38.3) in R, with functional enrichment analyzed via ClusterProfiler (v4.4.4). Intestinal microbiome sequencing data were processed on the Meggie Biocloud Platform (v3.2), employing DADA2 for ASV inference and α diversity analysis. Finally, a series of statistical and visual analyses, including α diversity analysis, community composition analysis, species difference analysis, correlation analysis, phylogenetic analysis, and functional prediction analysis, were performed based on ASV representative sequences and abundance information.

To prevent contamination, all procedures were conducted in a sterilized clean room using nuclease-free reagents and autoclaved consumables. Negative controls (reagent blanks) were included in DNA extraction and PCR steps to monitor contamination, with no detectable amplification observed in controls. Intestinal samples were dissected using sterile tools, and extracted DNA was verified for purity (A260/A280 = 1.8–2.0) and integrity via agarose gel electrophoresis. PCR amplicons were purified to remove contaminants, and bioinformatic pipelines excluded low-quality sequences and known contaminants during data analysis, ensuring the reliability of microbial community profiles.

### 2.7. qRT-PCR Validation

Using qRT-PCR to verify the sequencing results of the transcriptome, we detected the expression of key genes responding to different habitats and analyzed the molecular mechanism of *H. tientsinensis* adaptation to different habitats. To determine mRNA expression levels of candidate immunity-related DEGs, qRT-PCR was performed using the same RNA samples that were employed to construct the RNA-Seq library described above. Synthesized cDNA was used as a template for qPCR-PCR. We selected 6 genes in each of the three groups and glyceraldehyde-3-phosphate dehydrogenase (GAPDH) as the internal reference genes (Table 1). The primers were designed with Primer Premier 5.0 and synthesized through General Biosystems (Chuzhou, China) [29]. Reactions were performed in a 25 μL reaction mixture containing 12.5 μL of 2 × SYBR qPCR Mix, 1 μL of forward and reverse primers, 1 μL of cDNA, and 10.5 μL of RNase-free H_2_O [29]. qRT-PCR was performed under the following conditions: 95 °C for 3 min, 40 cycles at 95 °C for 15 s, 60 °C for 15 s, and 72 °C for 25 s. The relative expression levels of each gene were determined by the 2^−△△CT^ method [30].

## 3. Results

### 3.1. Optimal Habitat and Experimental Duration for H. tientsinensis

In the course of the preliminary experiment, compared to the aquatic and terrestrial environments, *H. tientsinensis* showed an obvious burrowing tendency and had the best vitality in the semi-terrestrial environment. In aquatic and terrestrial environments, however, the crabs’ activity decreased, and mortality increased over time. This phenomenon preliminarily indicated that the semi-terrestrial environment might be a suitable habitat for the crab. Based on these observations, the treatment time for the formal experiment was determined to be 10 h.

In the preliminary experiment, 15 crabs (5 per habitat) were monitored for 24 h to assess vitality and mortality. The semi-terrestrial group exhibited significantly higher activity scores (2.8 ± 0.2) than the aquatic (1.5 ± 0.3, *p* = 0.002) and terrestrial groups (1.2 ± 0.4, *p* = 0.001, one-way ANOVA), with 0% mortality within 24 h. In contrast, the aquatic and terrestrial groups had 80% (4/5) and 100% (5/5) mortality by 24 h, respectively (Fisher’s exact test, *p* < 0.05 vs. semi-terrestrial). Based on these results, the formal experiment used a 10 h treatment duration to minimize mortality bias. During formal experimentation, mortality rates were 0% (semi-terrestrial, 0/9), 11.1% (aquatic, 1/9), and 22.2% (terrestrial, 2/9), with semi-terrestrial mortality significantly lower than terrestrial (*p* = 0.048, chi-square test). These data confirm the semi-terrestrial environment as the most viable habitat for *H. tientsinensis*, aligning with their natural burrowing behavior and physiological adaptation.

### 3.2. Transcriptome Sequencing and De Novo Assembly

All raw-sequence reads data have been deposited in the NCBI Sequence Read Archive (SRA) database under the accession numbers SRR28638978, SRR28638977, SRR28638976, SRR28638975, SRR28638974, SRR28638973, SRR28638972, SRR28638971, and SRR28638970. We performed Illumina sequencing on nine samples from each of the three experimental groups, generating raw sequence data, which were subjected to quality filtering. A total of 46.398 Gb of clean data was obtained, with an average volume of 5.155 Gb per sample and Q30 base percentages above 95.37%, while GC contents ranged between 44.41% and 46.77% (Table 2). We performed de novo assembly using Trinity on clean data, followed by optimization and evaluation of the resulting assembly. The assembly yielded 337,150 unigenes and 547,835 transcripts, with an average N50 length of 597 bp. Moreover, 223,660 (66.34%) unigenes had lengths ranging from 1 to 400 bp.

### 3.3. Functional Annotation and Classification of Unigenes

To obtain more detailed functional information on the genes, we performed gene functional annotation using six major databases. A total of 7250 (2.15%), 9330 (2.77%), 28,798 (8.54%), 8448 (2.51%), 1203 (0.36%), and 7299 (2.16%) genes were mapped to GO, KEGG, Nr, Pfam, String, and Swiss-Prot, respectively (Appendix A).

A total of 7250 unigenes have been annotated in the GO database. GO analysis classified unigenes into three major functional categories: biological processes (BP), cellular components (CC), and molecular functions (MF). Regarding BP, the top three most significantly enriched categories for unigenes were cellular processes (GO:0009987), metabolic processes (GO:0008152), and biological regulation (GO:0065007). In terms of CC, significant enrichment was observed for the cellular anatomical entity (GO:0110165) and protein-containing complex (GO:0032991). The results of MF category enrichment showed that the majority of unigenes were associated with binding (GO:0005488), catalytic activity (GO:0003824), and transporter activity (GO:0005215) (Appendix A). The KEGG pathway enrichment analysis allocated 20,117 unigenes into six categories, with 4723 unigenes associated with metabolism, 45 unigenes associated with the immune system, and 20 unigenes associated with environmental adaptation (Appendix A).

### 3.4. Identification and Enrichment Analysis of DEGs

An overall analysis of DEGs revealed significant differences between *H. tientsinensis* under different habitats. In comparisons between the aquatic and semi-land ecotypes, 252 DEGs were identified. Comparisons between the aquatic and terrestrial ecotypes resulted in the identification of 308 DEGs, while 177 DEGs were detected in comparisons between the terrestrial and semi-land ecotypes (Figure 1).

To gain insight into the functions of DEGs, we performed GO functional annotation analysis. GO terms associated with osmotic regulation, such as response to stimulus (GO:0050896), were also enriched. As were several metabolic-process-related terms, including metabolic processes (GO:0008152) and catalytic activity (GO:0003824). Notably, the GO term “cellular anatomical entity” (GO:0110165) related to cytoskeletal proteins was significantly enriched in all three comparison groups. In addition, we performed a KEGG pathway enrichment analysis on the DEGs. In comparisons between aquatic and semi-land ecotypes, the caffeine metabolism (ko00232) and peroxisome pathways (ko04146) were significantly enriched. Among comparisons between aquatic and terrestrial ecotypes, the pyruvate metabolism (ko00620) and glyoxylate and dicarboxylate metabolism pathways (ko00630) were the top two most enriched pathways. Finally, among comparisons between terrestrial and semi-land ecotypes, the most significantly enriched pathways were valine, leucine, and isoleucine degradation (ko00280) and carbon metabolism (ko01200).

According to the enrichment of GO and KEGG pathways and the results of manual retrieval, we screened out some key DEGs. Some important genes related to metabolism were differentially expressed in the comparison of the three groups; for example, isocitrate dehydrogenase 3 (IDH3) was significantly upregulated in the aquatic and semi-terrestrial comparisons. Glycine N-methyltransferase (GNMT) and thetaiotaomicron diaminopimelate decarboxylase (LYSA) were significantly upregulated in aquatic and terrestrial comparisons. Solute carrier family 6 member 3 (SLC6A3) and solute carrier family 27 member 1 (SLC27A1-4) genes related to water regulation and ion transport were also significantly differentially expressed. There are also genes involved in cytoskeleton and motor regulation, such as dystonin (DST) and spastin (SPAST) (Figure 2).

### 3.5. Intestinal Microbial Composition and Diversity of H. tientsinensis in Different Habitats

The dilution curve constructed according to the alpha diversity index showed that, compared to data with less sequencing depth, the obtained sequencing data tended to be flat. This indicated that the current sequencing depth was sufficient to better reflect the composition of intestinal microorganisms of *H. tientsinensis* (Appendix A).

A total of 467,209 sequences from 27 experimental crabs were processed through a series of quality-control steps, including filtering out low-quality reads, trimming adapter sequences, and correcting base-calling errors. These processed sequences, hereafter referred to as optimized sequences, were classified into 25 phyla. Specifically, 21 phyla were detected in the terrestrial group, 18 phyla in the semi-terrestrial group, and 16 phyla in the aquatic group, and 12 phyla were common to all three groups.

At the phylum level, Firmicutes, Proteobacteria, and Bacteroidetes were the top three most abundant phyla. At the genus level, 194 genera were identified in the terrestrial group, 170 genera in the semi-terrestrial group, and 134 genera in the aquatic group, and a total of 80 genera were shared among all three groups (Figure 3).

There were no significant differences in Ace (Figure 4a), Chao (Figure 4b), and Sobs (Figure 4c) diversity indices of intestinal microbiota in three different habitats (*p* > 0.05).

At the phylum level, Firmicutes (38.0%) accounted for the highest proportion of intestinal flora in the aquatic environment, followed by Proteobacteria (30.9%) and Bacteroidetes (22.3%). Firmicutes accounted for the highest proportion of intestinal flora (36.6%), followed by Bacteroidetes (26.8%) and Proteobacteria (20.4%) in the semi-terrestrial environment. In the terrestrial environment, Firmicutes (46.4%), Proteobacteria (31.5%), and Bacteroidetes (18.2%) accounted for the highest proportions (Figure 5). There was no significant difference in the level composition and abundance of intestinal microflora in the three different habitats.

At the genus level, in the aquatic environment, the top five most dominant bacterial genera were *Candidatus bacilloplasma* (27.5%), unclassified_c_Alphaproteobacteria (23.5%), *Dysgonomonas* (9.0%), unclassified_f_Desulfocapsaceae (6.7%), and *Marinifilum* (5.7%). The top five most dominant bacterial genera in the intestinal flora in the semi-continental habitat were, respectively, *Candidatus bacilloplasma* (24.3%), unclassified_c_Alphaproteobacteria (15.5%), *Dysgonomonas* (3.1%), unclassified_f_Desulfocapsaceae (12.1%), and unclassified_o_Bacteroidales (8.3%). The top five most dominant bacterial genera in intestinal flora in terrestrial habitats were *Candidatus bacilloplasma* (32.3%), unclassified_c_Alphaproteobacteria (19.9%), *Vibrio* (8.6%), *Dysgonomonas* (6.6%), and *Tyzzerella* (6.2%) (Figure 6).

There were significant differences in the composition and abundance of intestinal microbiota in different habitats. The abundance of *Pseudomonas* in aquatic habitats was significantly higher than that in other habitat groups (*p* < 0.05). The abundance of *Malaciobacter* and *norank_f_Xanthobacteraceae* in aquatic habitats was significantly higher than that in the semi-terrestrial habitat group (*p* < 0.05). The abundance of *Acetobacterium* and *norank_f_Vermiphilaceae* in aquatic habitats was significantly higher than that in the terrestrial habitat group (*p* < 0.05). The abundance of *Dietzia* in semi-terrestrial habitats was significantly higher than that in other habitat groups (*p* < 0.05) (Figure 7).

### 3.6. Validation of RNA-Seq Data by qRT-PCR

To validate the transcriptome sequencing results, six DEGs involved in osmotic regulation and energy metabolism were selected for further confirmation by qRT-PCR. With GAPDH serving as the reference gene, the expression levels of these DEGs are shown in Figure 8. Notably, the results of qRT-PCR were consistent with the Illumina sequencing results, suggesting that the RNA-Seq expression analysis was accurate and reliable.

## 4. Discussion

Semi-terrestrial habitats connect aquatic and terrestrial environments, creating a complex environment [31]. As one of the semi-terrestrial crabs, *H. tientsinensis* can adapt to different habitats [32]. However, the molecular mechanisms underlying the adaptation of semi-terrestrial crabs to different habitats are still poorly understood. Therefore, in this study, we used RNA-seq to investigate the transcriptomic changes in *H. tientsinensis* in response to different habitats.

### 4.1. Cytoskeleton Protein and Motor Regulation

Water is considered the origin of life in the context of life evolution [33]. When organisms first transitioned to land, they faced gravitational loads without the buoyancy from the aquatic environment [34]. Different habitats impose varying gravitational loads on organisms [35]. Water’s density, about 800 times that of air, provides significant buoyancy [3]. Aquatic animals have more uniform gravitational loads due to buoyancy, while terrestrial animals encounter higher loads during locomotion [36]. This leads land crabs to have different muscle control patterns between land movement and swimming [36].

The cytoskeleton, a dynamic network of protein filaments, is crucial for various cellular functions like maintaining cell and organelle morphology and regulating cell adhesion, cell movement, and intracellular transport [37,38]. Genes related to the cytoskeleton play a key role in cell adaptation to different gravity environments and regulating cell motility. DST, encoding BPAG1 involved in cytoskeleton organization and intercellular interactions [39,40], was significantly upregulated in the AH group. Recessive mutations in the neuronal isoforms of DST, encoding dystonin, lead to abnormal actin cytoskeleton organization [41]. This indicates that *H. tientsinensis* regulates cell motility through upregulated DST in water compared to the semi-terrestrial environment. Tubulin genes are involved in crustacean cytoskeleton formation, cell division, and movement [42]. SPAST, which encodes spastin and maintains microtubule dynamic balance [43], was downregulated in both AH and TH comparisons. This suggests that changes in environmental gravity from semi-terrestrial to aquatic or terrestrial environments disrupt the dynamic balance of tubulin.

### 4.2. Water and Osmotic Pressure Regulation

Crustaceans typically maintain a water balance by regulating extracellular and intracellular ion concentrations [44]. Transcriptomic studies have shown that osmoregulation is a complex process involving multiple organs and signaling pathways, and many enzymes, as well as transporters, play an important role in maintaining osmotic pressure and ion balance [45,46]. The osmoregulatory strategy employed by the majority of marine crabs is to maintain internal osmotic pressure by the surrounding environment [47]. However, crabs living in estuarine areas with great changes in osmotic pressure can absorb or expel salt by adjusting their hyperosmolar or hypo-osmolar hemolymph [48].

It has been found that Ca^2+^-ATPase play an important role in the regulation of osmotic pressure in the gill tissue of *Epinehelus moara* [49,50]. ATPase sarcoplasmic reticulum Ca^2+^ transporting 1 (ATP2A) encodes a calcium ion transport ATPase, which is found in intracellular pumps located in the sarcoplasmic or endoplasmic reticula of muscle cells [51]. As a major Ca^2+^-ATPase, it is responsible for cytoplasmic Ca^2+^ reuptake into the sarcoplasmic reticulum, thereby affecting the osmotic regulation of cells [52]. In this study, ATP2A was upregulated in the AH group, affecting intracellular and intracellular Ca^2+^ concentrations in calcium signaling pathways, thereby regulating osmotic pressure. ATPase H+ transporting V1 subunit C1 (ATP6C) encodes a component of vacuolar ATPase (V-ATPase), a multisubunit enzyme that mediates the acidification of intracellular compartments of eukaryotic cells [53]. V-ATPase participates in acidifying and maintaining the pH of intracellular compartments and, in some cell types, is targeted to the plasma membrane, where it regulates acidification of the extracellular environment [54]. However, it is downregulated in the AH. We hypothesize that in aquatic environments, *H. tientsinensis* mainly regulates ion transporters in response to osmotic pressure changes in different environments.

The downregulation of ATP6C (encoding a V-ATPase subunit) in aquatic environments (AH group) likely reflects a reduced reliance on vacuolar acidification for osmotic regulation, as the surrounding water environment may provide more stable ion gradients compared to semi-terrestrial/terrestrial habitats. V-ATPases typically acidify intracellular compartments or the extracellular environment, but in well-buffered aquatic conditions, *H. tientsinensis* may prioritize other ion transporters like ATP2A (upregulated in AH), which mediates cytosolic Ca^2+^ reuptake and directly influences osmotic signaling. This shift aligns with findings in other crustaceans: for example, in the marine fish Epinehelus moara, Ca^2+^-ATPases (homologous to ATP2A) are critical for gill-mediated osmoregulation under salinity stress, while V-ATPase activity is upregulated in species facing extreme acid-base fluctuations (e.g., intertidal crabs). However, species-specific differences exist: semi-terrestrial crabs often exhibit enhanced ATP2A-dependent Ca^2+^ signaling to manage rapid osmotic shifts, whereas fully aquatic species may retain stronger ATP6C-mediated acidification for long-term ion balance. These divergent strategies highlight the adaptive plasticity of osmoregulatory genes, with *H. tientsinensis* optimizing energy allocation between acidification (via ATP6C) and calcium transport (via ATP2A) based on habitat water availability and ionic composition.

The SLC6 gene family participates in several physiological processes, including neurotransmitter signal transmission, amino acid transport, and homeostasis of osmotic pressure [48,55]. Studies have shown that members of the SLC6 gene family also play an important role in osmotic pressure regulation and salinity adaptation in aquatic animals, such as in *Oreochromis mossambicus* and *Danio rerio* [56]. SLC6A3 was upregulated in both the AH and TH groups. It suggested that SLC6A3 might have a significant impact on the osmotic pressure regulation of *H. tientsinensis*. When the external environment changes, SLC6A3 might be the crucial gene to help *H. tientsinensis* adapt to the change.

### 4.3. Energy Metabolism

In water-deficient environments, some animals may reduce their metabolic rate to decrease water loss. Crabs regulate osmotic pressure and acid-base balance through ion transporters and ion transport channels, which require a great deal of energy [57]. IDH3 is a key enzyme in the mitochondrial tricarboxylic acid (TCA) cycle, which catalyzes the decarboxylation of isocitrate into α-ketoglutarate and concurrently converts NAD+ into NADH [58]. This gene was also upregulated in the AH group, suggesting that the metabolic capacity of *H. tientsinensis* was improved in the aquatic environment compared with the semi-terrestrial environment. Moreover, cytochrome c oxidase (COX), the terminal enzyme of the mitochondrial respiratory chain, catalyzes the electron transfer from reduced cytochrome c to oxygen. Several studies have shown that COX plays an important role in the oxidative phosphorylation metabolism of shrimps and crabs. Many studies have demonstrated that COX2 may be a key gene in energy metabolism-related pathways in mice [59,60]. In our study, most DEGs related to energy metabolism, such as COX6 and Acyl-CoA dehydrogenase short chain (ACADS), were downregulated in TH. The downregulation of COX6 and ACADS in terrestrial environments likely reflects an energy conservation strategy beyond water balance, linked to reduced metabolic demands in drier, hypoxic conditions. As key players in mitochondrial respiration (COX6) and fatty acid oxidation (ACADS), their reduced expression would decrease oxidative phosphorylation and energy production, aligning with lower physical activity and oxygen availability in terrestrial habitats. This adaptive response minimizes energy expenditure under limited water and oxygen, a common strategy in arid-adapted crustaceans to sustain basal metabolism in resource-scarce environments, highlighting integrated metabolic and osmoregulatory adaptations in *H. tientsinensis*. This indicates that under relatively water-deficient terrestrial conditions, *H. tientsinensis* maintained its internal water balance by reducing its metabolic rate.

### 4.4. Intestinal Microbes

Intestinal microorganisms are critical for animal metabolism and immune regulation [61], with alpha diversity indices (e.g., Ace, Chao, Sobs) reflecting microbiota abundance and composition [51]. In *H. tientsinensis*, gut microbial alpha diversity remained stable across aquatic, semi-terrestrial, and terrestrial habitats (*p* > 0.05), indicating a consistent richness and community structure despite environmental shifts. This stability supports metabolic flexibility by maintaining a conserved core microbiome—dominated by Firmicutes, Proteobacteria, and Bacteroidota—that ensures baseline functions like carbohydrate/protein digestion and energy acquisition across habitats. Concurrently, genus-level compositional changes (e.g., Dietzia enrichment in semi-terrestrial environments) enable adaptive responses to niche-specific challenges, such as enhanced nutrient utilization or stress resistance. The balance of stable diversity and dynamic genus-level adjustments allows reliable basic metabolism alongside flexible functional modifications, facilitating the crab’s adaptation to heterogeneous aquatic–terrestrial interfaces.

At the phylum level, Firmicutes, Proteobacteria, and Bacteroidota were the dominant microbial phyla across different habitats. These three groups are important for host metabolism [62]. Firmicutes bacteria can aid food digestion [63], Proteobacteria engage in various metabolic processes like sulfur, methane, and hydrogen oxidation, sulfate reduction, and denitrification [64], and Bacteroides contribute to carbohydrate metabolism, supplying energy for the host [65]. These phyla are common in aquatic invertebrates’ intestinal microbiota. For instance, in different feeding modes of E. sinensis, similar phyla dominate [66,67,68], and in *S. paramamosain*, Firmicutes, Proteobacteria, Bacteroidota, and *Campylobacter* are dominant [65]. In aquatic environments, *Pseudomonas* and *Malaciobacter* in the *H. tientsinensis* gut, while opportunistic pathogens, also support metabolic adaptation. As dominant Proteobacteria, they aid nutrient cycling (e.g., organic matter utilization) and energy metabolism (e.g., fatty acid oxidation), enhancing host adaptation to water-rich habitats. Their presence reflects a balance between potential pathogenicity and beneficial symbiosis, leveraging microbial functions to improve environmental fitness. In our study, no significant differences in intestinal microbiota were detected at the phylum level, indicating that *H. tientsinensis* can adapt well within a certain range.

At the genus level, *Pseudomonas* and *Malaciobacter* abundances were significantly higher in the aquatic habitat (*p* < 0.05). These are common conditioned pathogens in aquatic animals’ intestines, being obligatory Gram-negative aerobes [69]. *Acetobacterium* abundance was also higher in the aquatic habitat compared to the terrestrial one (*p* < 0.05), potentially enhancing the gut’s reduced acetyl production, which could harm the crab’s gut and immune system. In semi-terrestrial habitats, *Dietzia* abundance was significantly higher (*p* < 0.05). As a Gram-positive aerobic bacterium, *Dietzia* can degrade hydrocarbons, providing growth-related nutrients [70] and being useful for cold-environment bioremediation [71]. These results imply that the terrestrial adaptation of *H. tientsinensis* may require more energy, and microbial composition changes could form an adaptation strategy.

## 5. Conclusions

In this study, we combined transcriptomic and microbiome techniques to systematically investigate the terrestrial adaptation mechanisms of *H. tientsinensis* for the first time. The results indicated that *H. tientsinensis* could adapt to both aquatic and terrestrial habitats. It was found that *H. tientsinensis* could adapt to terrestrial habitats by regulating energy metabolism and water balance in vivo with more complete protein quality control. In addition, intestinal microbes played an important role in the terrestrial adaptive leaves of the crab, helping it to better deal with the complex habitat. The results of this study lay a foundation for the exploration of the terrestrial adaptation mechanism of crabs and provide important information for the evolution of crustaceans from the sea to the land. This study’s genetic and microbial findings inform *H. tientsinensis* aquaculture by linking adaptive traits to breeding and disease prevention. Key genes (DST, ATP2A, and IDH3) serve as markers for selecting resilient strains, improving survival in fluctuating environments. Habitat-specific microbes (e.g., *Dietzia* in semi-terrestrial guts and *Pseudomonas* in aquatic) enable probiotic interventions to enhance metabolism/immunity and reduce pathogen risks. Integrating genetic selection with microbial management optimizes crab health and adaptability for commercial farming across diverse conditions.

## Figures and Tables

**Figure 1 animals-15-01244-f001:**
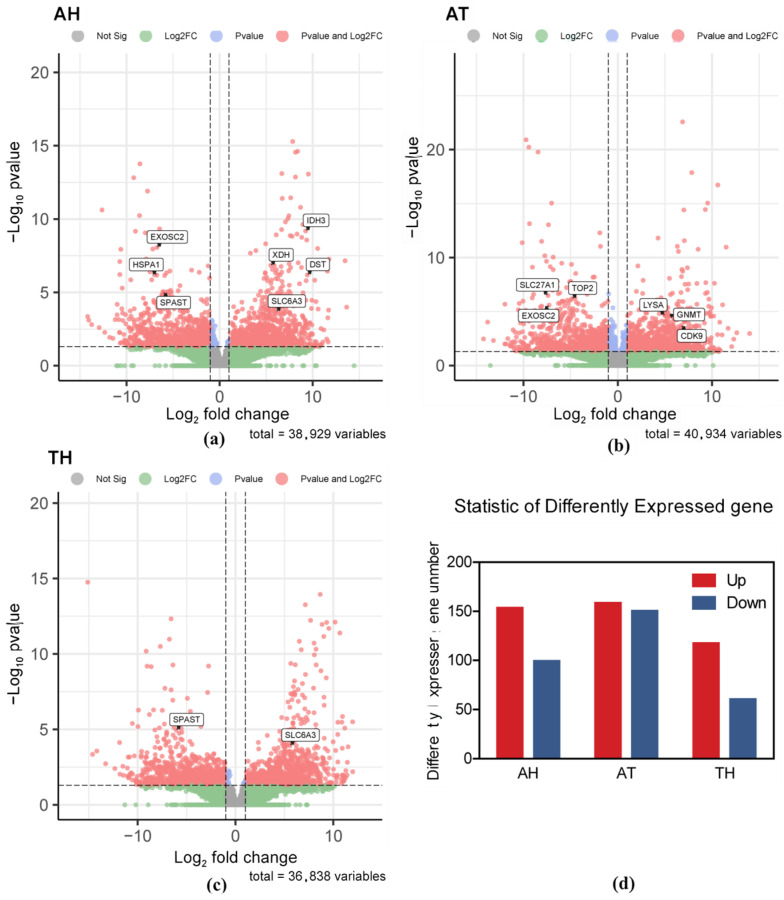
The red point represents significant upregulated differentially expressed genes (DEGs), and the blue point represents significant downregulated DEGs. Group AH is for the comparison between the aquatic and semi-terrestrial habitats; Group AT is for the comparison between the aquatic and terrestrial habitats; Group TH is for the comparison between the terrestrial and semi-terrestrial habitats. (**a**) Scatter plots of mean logarithmic values for gene expression in the AH; (**b**) scatter plots of mean logarithmic values for gene expression in the AT; (**c**) scatter plots of mean logarithmic values for gene expression in the TH; (**d**) statistics of differently expressed genes.

**Figure 2 animals-15-01244-f002:**
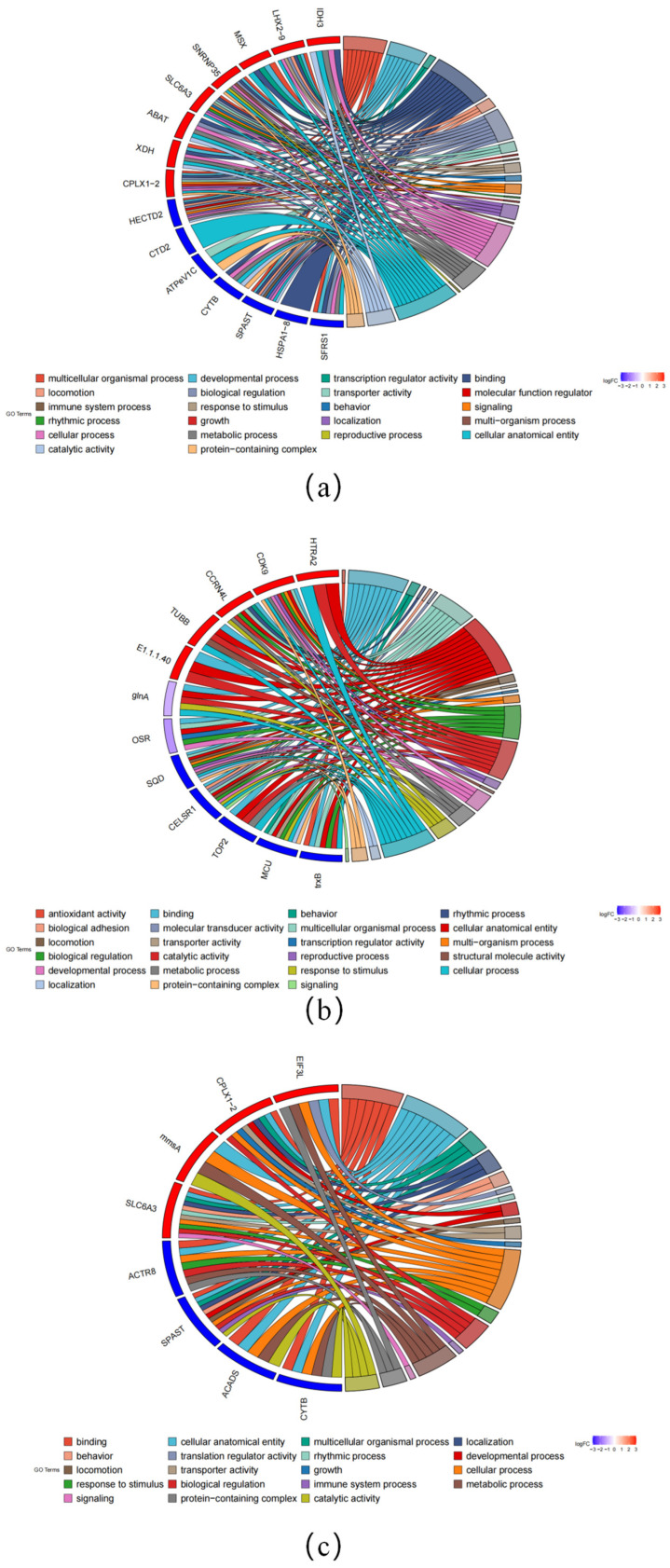
Annotated features of DEGs in GO: (**a**) DEGs of AH in the GO; (**b**) DEGs of AT in the GO; (**c**) DEGs of TH in the GO.

**Figure 3 animals-15-01244-f003:**
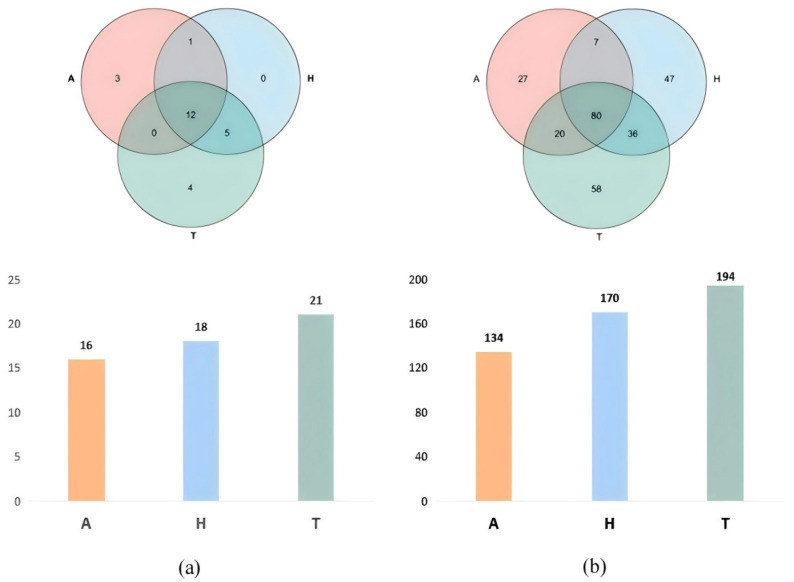
The intestinal microbiota of *H. tientsinensis* in different habitats at the phylum (**a**) and genus (**b**) levels.

**Figure 4 animals-15-01244-f004:**
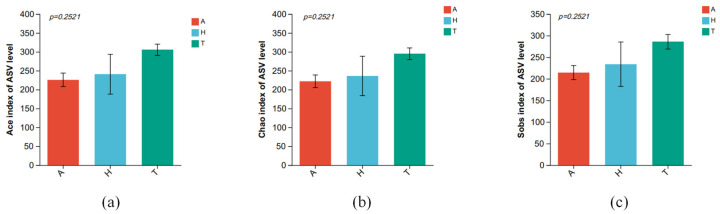
The difference in intestinal microbiota alpha diversity index in different habitats. (**a**) Ace, (**b**) Chao, and (**c**) Sobs.

**Figure 5 animals-15-01244-f005:**
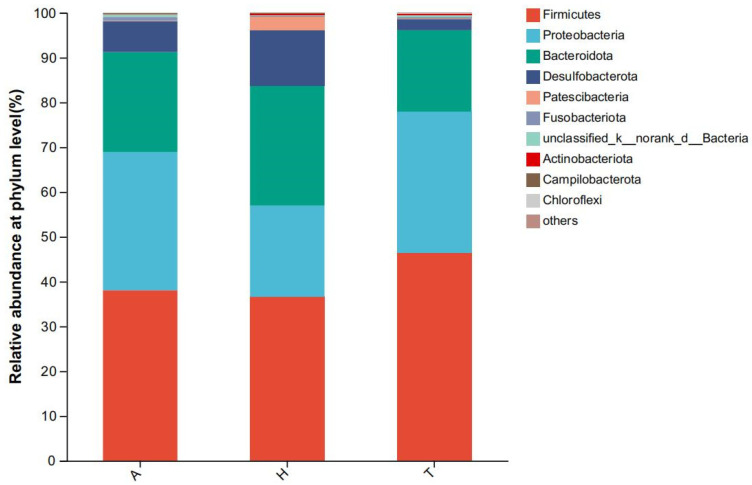
Community of intestinal microflora at the phylum level in different habitats.

**Figure 6 animals-15-01244-f006:**
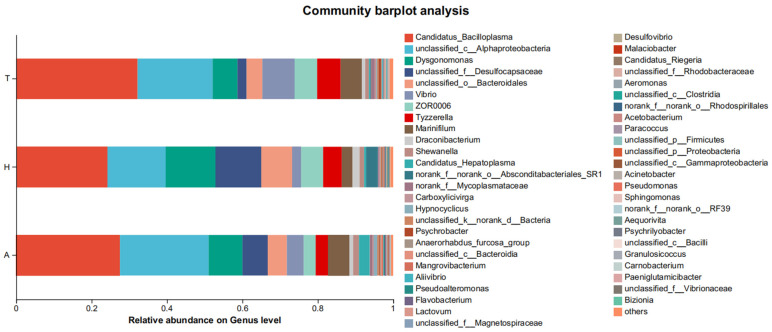
Community of intestinal microbiota at the genus level in different habitats.

**Figure 7 animals-15-01244-f007:**
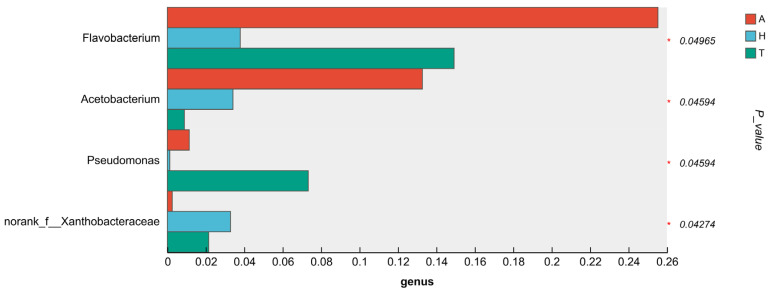
Differences in the levels of intestinal microbial flora in the analysis, with “*” representing a significant difference (*p* < 0.05).

**Figure 8 animals-15-01244-f008:**
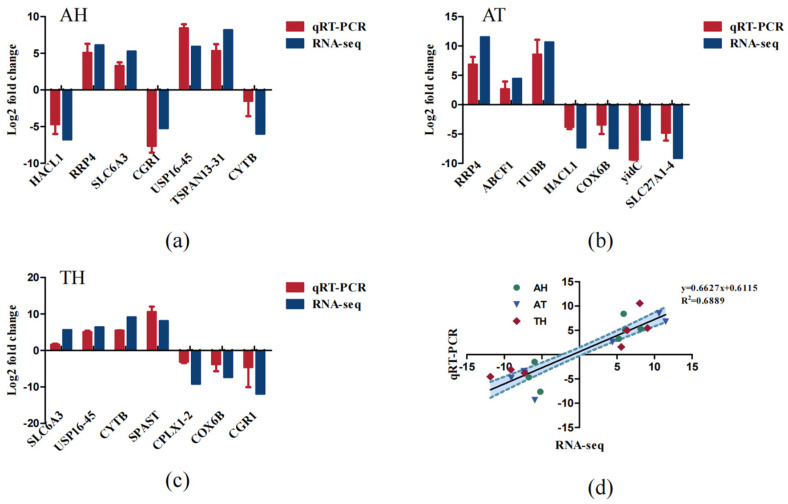
(**a**) Comparison of RNA-Seq and qRT-PCR expression in the AH; (**b**) comparison of RNA-Seq and qRT-PCR expression in the AT; (**c**) comparison of RNA-Seq and qRT-PCR expression in the TH; (**d**) comparison of RNA-Seq and qRT-PCR expression profiles in the gills of *H. tientsinensis*.

**Table 1 animals-15-01244-t001:** The primers of *H. tientsinensis* for qRT-PCR.

Gene Name	Forward Primer Sequence (5′-3′)	Reverse Primer Sequence (5′-3′)
*HACL1*	GGCGTCAATGCGTTATGTC	CCTTCCACCTCCTCCTCTG
*RRP4*	CAATAGTTGGGCGTATCACC	CCTAATCAAGCCATCATCAC
*SLC6A3*	TAGCGTGGAGTCTTCGGTTCT	GAGGAGGAGTAGGAGGTGTTGG
*CGR1*	AACTCCTCAAAGTCATCCCA	CCTCTATCGCAGCACTCATC
*USP16-45*	GAGCAACGCAAGCAAACAGA	CCCAGGGAGATTGAGGAGGT
*TSPAN13_31*	CCCTCAAACCCTCACTGCTA	TGCTGTTGTTGGTGCTCCTT
*CYTB*	CCCTGCTAATCCTCTTGT	CTCGGCTACTTCGTTCAT
*COX6B*	ACTCCCTACATTCACAGATTC	ACTTCAGAACAGCCTACACC
*yidC*	CTTGTCTCCTGTGACCTCCCG	CACCAGCATTCCTCGCCTC
*ABCF1*	AGAAGGACGAGGACGAAAG	TCCAAATCAATGCCGAAGT
*SLC27A1-4*	GGGGCTATGTGAGCAAAGGT	GATGCGGATGAAGCGAGGTA
*TUBB*	ATTTGCCTCCTGAGTTGTTC	CGTCATTATCGCTACTGCCTA
*SPAST*	GTGATACGCTGACTACGGG	GTGTAGGCTGCTGAACGAC
*CPLX1-2*	GGTGTAACCCAACTTAATCG	CAAATGTCATACCCTCAGAAT
*GAPDH*	ACCTGATGCTCCGATGTTT	CTTGTCCTGGTTGACTCCC

**Table 2 animals-15-01244-t002:** Basic statistics of sequencing.

Group	Total Raw Reads (*n*)	Total Clean Reads (*n*)	Q30%	GC%
A1-S	37,301,190	36,317,604	95.85	45.86
A2-S	44,511,258	43,564,496	95.90	45.11
A3-S	41,005,450	39,844,232	95.37	46.77
H1-S	43,072,134	42,012,372	96.00	45.99
H2-S	39,228,176	38,125,102	95.82	45.46
H3-S	38,576,812	37,781,692	95.84	44.41
T1-S	38,161,214	37,297,652	95.79	46.03
T2-S	41,179,406	40,285,070	96.01	46.14
T3-S	37,966,558	37,019,366	95.79	45.45
Assembly				
Number of total unigenes (*n*)		337,150		
Number of total transcripts (*n*)		547,835		
Average unigene length (bp)		520		
Average transcript length (bp)		700		
N50 of unigenes (bp)		597		
N50 of transcripts (bp)		1110		
N90 of unigenes (bp)		244		
N90 of transcripts (bp)		279		
Largest unigenes (bp)		33,403		
Smallest unigenes (bp)		201		

## Data Availability

All data and materials, as well as software applications, support their published claims and comply with field standards. The transcriptome data are available in the NCBI SRA (accession number: PRJNA1099413).

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
