# Peer review of "Comparative Transcriptomics and Intestinal Microbiome Analysis Provide Insights into the Semi-Terrestrial Adaptation of Helice tientsinensis"

_animals, 2025, doi:10.3390/ani15091244_

Round 1
Reviewer 1 Report
Comments and Suggestions for Authors
I think this research was conducted well. The authors isolated intestines/ biological material of crabs to analyze variations of genetics expressions of crabs in different environments and then presented the results. Although I am not too familiar with biotechnological methods, the methods and results seemed clear and I was able to research and understand them. For the discussion, I would revise the first paragraphs of each section to integrate more with your research. In my opinion, the first paragraph of the discussion should highlight the major results and/or points of interest regarding the results. Other than that, the English was great with a few minor exceptions.
Reviewer 2 Report
Comments and Suggestions for Authors
Review for the paper “Comparative transcriptomics and intestinal microbiome analysis provide insights into the semi-terrestrial adaptation of Helice tientsinensis” by Zhengfei Wang and co-authors submitted to “Animals”.
The authors of this research paper conducted an analysis of the terrestrial adaptations of the grapsid crab Helice tientsinensis. These crabs are particularly known for their existence in intertidal zones, where they engage in extensive burrowing and exhibit strong adaptability to semi-terrestrial environments. This species serves as an important model for studying the evolutionary mechanisms underlying terrestrial adaptation in crabs, in addition to its significance in Chinese aquaculture. Their analysis identified critical genes associated with the regulation of cytoskeleton motor function, management of water and osmotic pressure, and adaptations in energy metabolism. Furthermore, the authors examined the intestinal microbial communities associated with H. tientsinensis across different environmental contexts. They found habitat-specific differences in the composition of these microbial communities, suggesting that the gut microbiome plays a significant role in the crab's ability to adapt to land. The results of this study may have important implications for broader research into crustacean evolution and adaptation.
The paper is well organized and written; however, certain important details regarding the methodology are missing. A broader discussion is also needed. Therefore, the following recommendations are offered to further improve the paper.
Recommendations:
Abstract.
L 30-32. Change “Regarding intestinal microbiota, there were no significant differences in alpha diversity among habitats, but differences at the genus level” by “Regarding the gut microbiota, no significant differences in alpha diversity were found between habitats, but there were differences at the genus level”.
Introduction.
L 50. More information is needed on the importance of this crab in aquaculture, including annual catch rates, export values, if any, and other relevant data.
L 51. The authors should report the unique morphological or physiological features of H. tientsinensis that make it an ideal model species. What distinguishes it from other Grapsidae species in terms of terrestrial adaptation?
Material and Methods.
L 89-90. The study period should be included in the text. A map of the study area should also be provided.
L 98-105. More details about this preliminary experiment should be included. How many crabs were used? Were their size and sex similar to those used in the main experiments?
L 107. The authors should provide more information about the animals used (size, age, sex). What criteria were used to select the 27 H. tientsinensis individuals? What measures were taken to control for individual variation?
L 112. How many samples were used for the RNA analysis? Why was gill tissue specifically chosen for RNA extraction, and were other tissues (besides gut tissue) considered for molecular analysis to investigate terrestrial adaptation?
L 144. The authors should report the criteria used to define differentially expressed genes (DEGs). Were specific thresholds for fold change and statistical significance (e.g., log2FC > 2 and p-value < 0.05) used?
L 155. The authors should describe the measures taken to ensure that no contamination occurred during the extraction of microbial genomic DNA.
L 166. The authors should provide information on the software used for these analyses.
Results.
Section 3.1. This section is descriptive. The authors should provide statistical information to support the best vitality of the crabs in the semi-terrestrial environment. What was the mortality rate during the experiment?
L 209. Table S1 is referenced but not included in the paper.
L 218. Figure S1 is cited but not included in the paper.
L 221. Figure S2 is cited but not used in the paper.
Figure 1. The authors should define all abbreviations in the caption.
Figure 2 is difficult to understand due to very small font size.
L 264. Figure S3 is cited but not included in the paper.
Figure 3. The authors mention differences in microbiota composition. However, it remains unclear whether the 10-h experimental period is sufficient to achieve such differences.
L 279, 288, 305, 307, 309, 310. What test was used to examine the data for possible differences between groups?
L 292. “Generic level” should be changed to “genus level”.
Figure 6. The font size should be increased.
Discussion.
L 347. The authors should discuss the biological significance of the DST gene's up-regulation in water. Does this suggest a specific cellular reorganization process that aids in motility or structural changes in an aquatic environment?
L 372. The authors should discuss more clearly the physiological implications of the ATP6C down-regulation in aquatic environments. Could it reflect a reduced role of acidification in maintaining osmotic balance due to the surrounding water environment? Are the roles of ATP2A and ATP6C consistent across other semi-terrestrial or aquatic crustaceans? Are there known species-specific differences in the importance of these genes for osmoregulation?
L 400-402. Why might COX6 and ACADS be downregulated in terrestrial environments beyond conserving water balance? Could this reduction also be linked to lower physical activity or metabolic demands in drier conditions?
L 410. The authors found that the alpha diversity index of the intestinal microbiota remains stable across habitats. How does this stability contribute to the metabolic flexibility of H. tientsinensis?
L 422-424. The authors should discuss the ecological roles of Pseudomonas and Malaciobacter in the gut of H. tientsinensis in aquatic environments. Are they strictly pathogens, or could they also contribute to certain beneficial processes (e.g., nutrient cycling or immune modulation)?
How are practical implications for aquaculture, such as breeding strategies or disease prevention, connected to the genetic and microbial findings in this study?
Round 2
Reviewer 2 Report
Comments and Suggestions for Authors
The authors have sufficiently adreesed all my concerns.
The supplementary material is cited but not included.